# Mapping Tobacco Fields Using UAV RGB Images

**DOI:** 10.3390/s19081791

**Published:** 2019-04-15

**Authors:** Xiufang Zhu, Guofeng Xiao, Ping Wen, Jinshui Zhang, Chenyao Hou

**Affiliations:** 1State Key Laboratory of Earth Surface Processes and Resource Ecology, Beijing Normal University, Beijing 100875, China; zhuxiufang@bnu.edu.cn (X.Z.); zhangjs@bnu.edu.cn (J.Z.); 2Key Laboratory of Environmental Change and Natural Disaster, Ministry of Education, Beijing Normal University, Beijing 100875, China; 3Institute of Remote Sensing Science and Engineering, Faculty of Geographical Science, Beijing Normal University, Beijing 100875, China; houcy@mail.bnu.edu.cn; 4Powerchina Kunming Engineering Corporation Limited, Kunming 650051, China; 13683177203@163.com

**Keywords:** tobacco field, UAV image, morphology, convolution

## Abstract

Tobacco planting information is an important part of tobacco production management. Unmanned aerial vehicle (UAV) remote sensing systems have become a popular topic worldwide because they are mobile, rapid and economic. In this paper, an automatic identification method for tobacco fields based on UAV images is developed by combining supervised classifications with image morphological operations, and this method was used in the Yunnan Province, which is the top province for tobacco planting in China. The results show that the produce accuracy, user accuracy, and overall accuracy of tobacco field identification using the method proposed in this paper are 92.59%, 96.61% and 95.93%, respectively. The method proposed in this paper has the advantages of automation, flow process, high accuracy and easy operation, but the ground sampling distance (GSD) of the UAV image has an effect on the accuracy of the proposed method. When the image GSD was reduced to 1 m, the overall accuracy decreased by approximately 10%. To solve this problem, we further introduced the convolution method into the proposed method, which can ensure the recognition accuracy of tobacco field is above 90% when GSD is less than or equal to 1 m. Some other potential improvements of methods for mapping tobacco fields were also discussed in this paper.

## 1. Introduction

China is the top tobacco producer and consumer in the world [1]. According to statistics, the tobacco-sown area in China was 1.273 million hectares in 2016, of which 1.206 million hectares were flue-cured tobacco [2]. Flue-cured tobacco was the most popular type of tobacco planted in the world, and it is the main raw material of the cigarette industry. Revenue from tobacco enterprises and from the tobacco leaf tax in China was 37.293 billion CNY (about 5.810 billion USD) and 13.054 billion CNY (approximately 2.034 billion USD) in 2016, respectively [2]. The estimation of the tobacco planting area can help improve tobacco planting, improve the supervision of the scale of tobacco growers, and elucidate the yield and supply of tobacco. Tobacco planting information is an important basis for tobacco production management, and it can provide decision-making support for tobacco planting and management. It is also important economic information for national macro management and decision-making. In addition, insurance companies also need to understand the scope of the spatial distribution of tobacco in order to achieve accurate insurance and claim estimates and to reduce the moral hazard problems in traditional contract claims.

The routine monitoring of the tobacco planting area depends on the method of ground investigation, which is time-consuming and costly. It can only be verified by local measurements and can easily be affected by human factors; real-time and accurate planting area data cannot be obtained at a large scale. Remote sensing technology has been widely used in planting area monitoring of major crops because of its large coverage area, quickly accessible information, strong real-time information, no limit on ground conditions, and low cost compared with artificial ground surveys [3,4,5,6,7]. However, there are relatively few studies on the extraction of tobacco planting information using remote sensing data [8,9,10,11,12].

Compared with major crops, tobacco fields are fragmented and dispersed. During the growing season of tobacco, there are also many other types of crops growing, which can easily cause misclassifications. Thus, the requirements for the spatial and temporal resolution of the remote sensing data for identifying tobacco are very high. However, the traditional visible remote sensing data are often affected by cloud and rain weather, while the resolution of radar data is usually quite low. Both of these methods cannot easily meet the needs of real-time monitoring of the spatial distribution of tobacco.

Unmanned aerial vehicle (UAV) remote sensing systems have become a popular topic in the world because they are mobile, rapid and economic [13]. In recent years, with the rapid progress of UAV remote sensing technology, it has gradually advanced from being used solely in research to use in practical applications, including precision agriculture [14,15,16,17], vegetated area monitoring [18,19,20], crop area mapping [21,22], environment monitoring [23,24], wildlife research [25,26], archaeological application [27], etc. Clear tobacco field images with a high spatial resolution at the centimetre level can be obtained by using a UAV. In addition, with a UAV, it is easier to ensure that high quality data are acquired at the best monitoring phase in practical applications due to its flexible flight time. However, to our knowledge, there are only a few studies on tobacco field mapping based on UAV images.

In this paper, the Yunnan Province is taken as an example, as it is the top province for tobacco planting in China. Multispectral data are often unavailable in cloudy and rainy areas such as Yunnan province. It is scarcely possible to obtain multispectral data received at the best monitoring period and covering all planting areas of Yunnan province. The optimal monitoring phase of the tobacco field is analysed, UAV Red-Green-Blue (RGB) images are acquired at the optimal monitoring phase, and an automatic and high-precision identification method for tobacco fields based on UAV RGB images is developed. This study contributes to understanding effective monitoring of the tobacco planting area and distribution range, provides a reference for the extraction of tobacco planting information in other regions, and also contributes to the extraction of plastic film farmland.

## 2. Study Area and Data

The present study was conducted in Ludian County of Zhaotong, Yunnan Province, China, which is the top producing county for high-quality flue-cured tobacco (Figure 1). The annual average temperature of this city is 12.1 degrees, the frost-free period is 220 days, and the annual precipitation is 900 ms. Of the total area in Ludian County, 87.9% is mountainous. The main crops produced in this area include winter wheat, spring maize, summer maize, medium rice and flue-cured tobacco. The specific phenological records are shown in Table 1. Crops show different spectral characteristics at different stages of growth and development, and this seasonal rhythm varies between different crop types. Analyzing the phenological characteristics of flue-cured tobacco and other crops and selecting the best phase for flue-cured tobacco identification are helpful to lessen the interference from related factors and improve the accuracy of flue-cured tobacco identification.

Tobacco belongs to the Solanaceae family of annual crops. Its growth period can be divided into two parts: seedbed and field. The seedbed period is from tobacco planting to transplanting, including the emergence, cross, rooting and seedling stages. The field period is from transplanting to harvesting, including the returning seedling stage, root elongation stage, flourishing stage and maturity stage (Figure 2). The transplantation of flue-cured tobacco in the study region is typically over in early May. The ground surface of the tobacco field was a mixture of vegetation, bare land and plastic film. Summer maize had not been sown and shows the spectrum of the bare soil. Winter wheat was mature and is represented as vegetation in the spectral information. Middle rice had just emerged, showing weak vegetation spectral information. Thus, summer maize, winter wheat and middle rice were not confused with flue-cured tobacco. Spring maize was covered with plastic film, but the film mulching time of spring maize is approximately one month earlier than that of flue-cured tobacco. In early May, spring maize is at the jointing stage. In addition, the planting methods of spring maize and flue-cured tobacco were different. Spring maize was planted in double rows and was interplanted with potatoes. The plastic film of the spring maize was white. Flue-cured tobacco was planted in single rows without interplanting and was covered with black film (Figure 3). Therefore, the returning seedling stage after transplantation was beneficial to distinguish flue-cured tobacco from the other crops. On one hand, all of the flue-cured tobacco has been transplanted, ensuring that tobacco fields will not be missed at this stage. On the other hand, at this stage, tobacco seedlings have just started to grow, spring maize is at the jointing stage, and the vegetation information of the spring maize is stronger than that of tobacco, which is beneficial to reduce commission errors.

Based on the above analysis, we carried out aerial photography on 4 May 2018 and obtained UAV RGB images of the study area. The UAV remote sensing system used in the experiment was the fixed wing electric UAV remote sensing platform, called DM-150 (which was equipped with an RGB camera (Sony ILCE-7R, Sony, Tokyo, Japan). The body mass is 10 kg, body diameter is 205 cm, maximum effective load is 6 kg, maximum flight speed is 150 km/h, endurance time is 3 h and maximum flight height is 5000 m. The sensor is 35 mm full frame (35.9 × 24 mm) Exmor CMOS (Complementary Metal-Oxide-Semiconductor) sensor (Tokyo, Japan). The lens is SONY Sonnar T* FE 35 mm F2.8 ZA (SEL35F28Z) (Tokyo, Japan) with the equivalent focal length of 35 mm. The camera captures images with 7360 × 4912 pixels resolution and the pixel size is 4.9 μm × 4.9 μm. The flight path was designed to ensure overlapping images of at least 30% side overlap and 70% forward overlap. UAV flight height is 285 m. The image was acquired with a ground sampling distance (GSD) of 0.04 m. The resulting image covered an area of 1,048,080 m^2^ and was georeferenced using 15 ground control points (GCPs). The acquired average positioning accuracy was 0.08 m. There were 400 images utilized for the orthophoto generation and Orthophotos were produced by using Pix4Dmapper software (V4.0, Swiss Federal Institute of Technology Lausanne (EPFL), Lausanne, Switzerland).

## 3. Methodology

In this paper, a method for extracting the tobacco planting area based on image morphology is presented in Figure 4. The main steps included making supervised classifications using the raw UAV images and extracting the mixed class of the black plastic film and the black buildings; using erosion and dilation algorithms of image morphology to remove the large area of buildings in the mixed class and obtain the preliminary distribution of the plastic film; using the erosion algorithm to eliminate the small area of fragmented spots from the plastic film distribution map; using the dilation algorithm to obtain the preliminary distribution of the tobacco fields; setting the area threshold to eliminate the noise spots and get the final flue-cured tobacco fields.Supervision classification: the types of land cover in the study area were first classified into nine categories: woodland, grassland, dirt road, facilities land, bare cultivated land, white building, black building, white plastic film (spring maize film) and black plastic film (tobacco film) by using the maximum likelihood classification algorithm. The training samples were selected by visual interpretation (Table 2).Extraction of the mixed class of black buildings and tobacco film: black buildings were easily confused with tobacco film; therefore, based on the land cover map produced in the first step, we recoded the tobacco film and the black buildings as 1 (refers to the mixed class of black building and black film), and, then, we recoded the other land cover types as 0.Elimination of the large area of black building land in the mixed class and extraction of the tobacco film distribution: by observing the image, we can see that the width of a single tobacco film is relatively narrow (usually less than 1 m), far less than the width of the buildings. Thus, we used the erosion and dilation algorithms of image morphology to remove the large area of black buildings in the mixed class. First, erosion was carried out. Assuming that the maximum width covered by a row of film is *B_max_*, the GSD of the image is *S*, a numerical upwarding operation is *T*, the core size of the erosion is *K*, the element value in the core is 1, and the erosion width is set as $S\times \frac{K+1}{2}$. *K* = 0 refers to when no process in the elimination of the large area of black buildings in the mixed class and extraction of the tobacco film distribution is made. After erosion, the tobacco film and other small spots in the mixed class were eliminated, and the edges of the black buildings were corroded, which resulted in the reduction of the scope of the black building land. Then, we carried out the dilation algorithm on the corroded black building land. The expanded core size is consistent with the erosion core size, and the element value in the core is set as 1. At this point, the distribution map of the large area of black buildings has been obtained. Then, we masked the large area of buildings from the binary map produced in the second step to obtain the distribution of the tobacco films that still included a small area of buildings (regarded as noise):(1)T=⌈BmaxS⌉,(2)$$K=\left\{ \begin{array}{ll} T & T\;\mathrm{is}\;\mathrm{an}\;\mathrm{odd}\;\mathrm{number},\\ T+1 & T\;\mathrm{is}\;\mathrm{an}\;\mathrm{even}\;\mathrm{number},\\ 0 & T<3. \end{array}\right.$$Elimination of the small area of fragmented spots from the tobacco film distribution map: after the previous operation, the binary map mainly contained the tobacco film and some fragmented spots. These fragmented spots are the commission errors because other land cover types were mistakenly classified as tobacco film or as black buildings during the process of supervised classification. We carried out erosion to remove them. Assuming that the minimum width covered by a row tobacco film is *B_min_*, the GSD of the image is *S*, and the core size of the erosion is $K^{\prime }$. $K^{\prime }$= 0 refers to when no process in the elimination of the small area of fragmented spots from the tobacco film distribution map is made. $T^{\prime }$ is a numerical upwarding operation, the element value in the core is 1, and the erosion width is set as $S\times \frac{K^{\prime }+1}{2}$. After erosion, the edges of the tobacco films were also corroded although the fragmented spots were removed. Thus, we carried out the dilation algorithm on the corroded tobacco films to restore the original tobacco film size. The expanded core size is consistent with the erosion core size, and the element value in the core is set as 1:(3)T′=⌈BminS⌉,(4)$$0\leq K^{\prime }<\left\{ \begin{array}{ll} T^{\prime } & T^{\prime }\;\mathrm{is}\;\mathrm{an}\;\mathrm{odd}\;\mathrm{number},\\ T^{\prime }-1 & T^{\prime }\;\mathrm{is}\;\mathrm{an}\;\mathrm{even}\;\mathrm{number}. \end{array}\right.$$Dilation of the tobacco films to get the preliminary distribution of the tobacco fields: assuming that the maximum spacing between adjacent films is *D_max_*, the GSD of the image is *S*, the size of the expanded core is $K^{\prime \prime }$, $T^{\prime \prime }$ is a numerical upwarding operation, and the element value in the core is 1. After the dilation operation, tobacco films were merged into tobacco fields:(5)T″=⌈DmaxS⌉,(6)$$K^{\prime \prime }=\left\{ \begin{array}{ll} T^{\prime \prime } & T^{\prime \prime }\;\mathrm{is}\;\mathrm{an}\;\mathrm{odd}\;\mathrm{number},\\ T^{\prime \prime }+1 & T^{\prime \prime }\;\mathrm{is}\;\mathrm{an}\;\mathrm{even}\;\mathrm{number},\\ 3 & T^{\prime \prime }<3. \end{array}\right.$$Generation of the final distribution of the tobacco fields: after a series of dilation and erosion operations, the land cover types on the binary map were only the flue-cured tobacco fields and some black building pixels that are not eliminated; however, the area of the black buildings was far smaller than the area of the flue-cured tobacco field. By setting an area threshold of the spots, the black building spots were removed, and the final distribution map of the flue-cured tobacco fields was obtained. In this paper, the area threshold was set as 200 m^2^.Accuracy validation: we digitalized the fields of flue-cured tobacco in the study area by visual interpretation, and we used them as a reference to assess the accuracy of the flue-cured tobacco distribution map produced by the automatic identification method proposed in this paper.

## 4. Results

Based on the 0.04 m UAV image (Figure 5a), the binary maps of the tobacco film (including some of the black buildings) and the non-tobacco film in the study area were extracted by a maximum likelihood classification (Figure 5b). The digital camera carried by the UAV is similar to the human visual system. The colour of the UAV image was based on RGB three primary colour imaging. The digital number (DN) values of the tobacco films and black buildings were quite similar, causing commission errors in the classification between the tobacco films and the black buildings. Therefore, in the binary map, one was a mixture of tobacco films and some black buildings. Based on this information, a series of image morphology erosion and dilation algorithms were applied to get the spatial distribution of the tobacco fields (Figure 5d). We digitalized the tobacco fields in the study area by visual interpretation and used them as a reference (Figure 5c) to assess the accuracy of the tobacco field distribution map produced by the automatic identification method proposed in this paper. The results show that the produce accuracy, user accuracy and overall accuracy of the automatic identification of the tobacco fields were 92.59%, 96.61% and 95.93%, respectively. The main error in the automatic identification of the tobacco fields was that some of the black buildings were mistakenly classified as part of the tobacco field (see the white rectangle indicator in the lower right corner of Figure 5d).

## 5. Discussion

### 5.1. The Influence of the GSD of the Image on Mapping Tobacco Fields

To analyse the effect of the GSD of the image on the method proposed in this paper, we generated a series of simulated images with GSDs ranging from 0.08 m to 1 m with 0.04 m as the interval based on the original 0.04 m UAV image by using an upscaling method. For each simulated image, the distribution map of the tobacco fields was created by the method proposed in this paper and using the same training dataset, and its accuracy was evaluated by using the same reference map, which was produced by a visual interpretation of the original 0.04 m UAV image. The results are shown in Table 3 and Table 4. The overall accuracy of tobacco field identification decreased as the UAV image GSD decreased. When the image GSD was reduced to 1 m, the overall accuracy decreased by approximately 10%. The reduction in tobacco field identification accuracy was mainly caused by a reduction in user accuracy. When the image GSD was reduced to 1 m, the user accuracy was 72.06%, which was approximately 22% lower than the user accuracy of the tobacco field identification with the 0.04 m image GSD (which was 96.61%). Unlike the change in user accuracy, the produce accuracy tended to increase with the reduction in image GSD. When the image GSD was reduced to 1 m, the produce accuracy was 97.39%, which was approximately 5% higher than the produce accuracy of the tobacco field identification with the 0.04 m image (which is 92.59%). With the reduction in image GSD, the user accuracy decreased, and the produce accuracy of the products increased; therefore, the commission error increased, and the omission error decreased. This phenomenon may be because, as the image GSD decreased, the film and the bare land in the tobacco fields gradually merged into mixed pixels. Hence, the difference between the DN values of the mixed pixels and the DN values of the black buildings and roads gradually decreased in the UAV image, causing the commission error to increase. In addition, when the image GSD was reduced to 0.20 m, the small area of fragmented spots merged into the surrounding pixels to form mixed pixels (Table 4); therefore, $K^{\prime }$ equalled 0, referring to no need to work with the fragmented spots and more. When the image GSD was reduced to 0.32 m, the tobacco film and the bare land pixels merged into mixed film pixels. The area of the spots composed by the mixed film pixels was nearly equivalent to the area of the spots composed by the black building pixels. It was impossible to remove the large black buildings by the morphological methods, and thus *K* equalled 0, meaning that no elimination of the large area of black buildings in binary map was conducted. 

### 5.2. Potential Improvements on the Proposed Tobacco Mapping Method 

When the image GSD was reduced to 0.32 m, both *K* and $K^{\prime }$ equalled 0, which meant that the third and fourth steps in technical flowchart (Figure 4) were not completed. Therefore, we only needed to carry out the dilation operation of the tobacco film pixels to form the tobacco fields. However, it was difficult to determine an appropriate size of the expanded nucleus. A too large expanded core will lead to several adjacent tobacco fields merging into one bigger tobacco field, while a too small expanded core will leave out some pixels in the tobacco fields (see Figure 6). Therefore, we tried to introduce the convolution method into the proposed morphological-based tobacco field extraction method to reduce the commission and omission errors.

With the reduction in UAV image GSD, the area ratio of the film mixed pixels in the tobacco field increased gradually. When the UAV image GSD was below 0.44 m, the number of the film pixels identified in the tobacco field was much higher than that of the bare land pixels. Therefore, before the fifth step in Figure 4, the median convolution operation was performed on the binary map first, and the size of the convolution window was set as (*B_max_*+ *D_max_*)/s, in which the meaning of *B_max_*, *D_max_* and _S_ is consistent with that in formulas 1–5. Since the pixel values in the window were only 0 (non-tobacco film) and 1 (tobacco film), after the operation of the median convolution, if the tobacco film pixels occupied a greater area ratio in the window, the value of the pixel in the centre of the window was set as 1; otherwise, it was set as 0. As we mentioned above, when the UAV GSD was below 0.44 m, the tobacco film pixels occupied more area in the tobacco field. Therefore, the non-tobacco film pixels in the tobacco field could be removed to reduce the omission error after using the median convolution operation on the binary map with an image GSD below 0.44 m. 

We combined convolution with the morphology method to extract the tobacco field from UAV images with GSD ranging from 0.48 m to 1 m, and we evaluated the extraction accuracy compared with the reference map. The results are shown in Table 3 and Table 4. After the convolution operation, the user accuracy improved, increasing on average by approximately 20.83%, but the producer accuracy decreased, decreasing on average by approximately 16.88%. The overall accuracy of tobacco field recognition improved by 4.76%, which was mainly dependent on the improvement in user accuracy. The improvement in user accuracy was mainly due to the reduction in the commission error between the tobacco field and the road (Figure 7d). When extracting tobacco fields based only on the morphology method, it was easy to expand too much, causing the roads between the tobacco fields to be mistakenly classified as part of the tobacco fields (Figure 7c). The reduction in produce accuracy was mainly due to the filtration of the broken tobacco film pixels on the edge of the tobacco field after the convolution operation, resulting in a certain omission error (Figure 7d).

### 5.3. Deficiencies in This Study and Future Improvements

The GSD of the UAV image has an effect on the recognition accuracy of tobacco fields. In this paper, the erosion and dilation core size are directly related to the UAV image GSD (see formulas 1 to 6). Excessive erosion will increase the omission error, while excessive dilation will increase the commission error. The simulation results show that, with a reduction in UAV image GSD, the overall accuracy of tobacco field recognition based on morphology decreases, especially when the GSD is lower than 0.44 m, and the overall accuracy is less than 90%. By combining convolution with morphology, the overall accuracy can be increased to over 90%, but the overall accuracy is still approximately 4.62% lower compared with the tobacco field recognition results based on the 0.04 m UAV image.

Image acquisition time has an effect on our proposed method. Crop fields show different spectral characteristics with crop growth, and selecting the data acquired in the optimal monitoring time, in which there is the biggest spectral differences between the tobacco field and other land cover types, benefits the recognition accuracy [10,28]. In this paper, we analysed the phenology of the main crops at the same period, and we determined that the optimal monitoring time in our study area was at the beginning of the seedling stage of tobacco, in which tobacco transplantation is finished and tobacco fields are covered by plastic film. UAV images were taken at the best monitoring time in the study area and used for testing our proposed method. However, no images from the whole growing season were collected for further analysis. With the growth of flue-cured tobacco, the tobacco seedlings will gradually become larger. Therefore, the vegetation information in the tobacco fields will increase gradually, and the information of the film and the bare land will decrease gradually. The applicability of the method proposed in this paper decreases as the tobacco grows. In the future, we need to collect time series data and analyse the effective time windows to provide a reference for determining the appropriate data acquisition time.

In the UAV RGB image, the DN value of the tobacco film is close to the DN value of the black buildings and roads. Hence, it is easy to misclassify them. Although the method proposed in this paper has taken certain considerations for this problem, it is difficult to distinguish the tobacco film from the buildings and roads effectively by the DN value of a single-phase UAV RGB image, which is the main source of error in the tobacco field identification. The following improvements could be considered in the future. If only single-phase UAV RGB images can be acquired, we could try to transform the DN value to different indices, or we could extract the different texture information of the image to enhance the difference between the tobacco films and the roads/buildings [29]. We also could try to use an object-oriented classification method to first extract building objects [30,31], remove the buildings from the UAV RGB images, and then use our proposed method to extract the tobacco fields. If multi-temporal UAV RGB images can be obtained, we could consider collecting two UAV RGB images before and after transplantation, and then we could identify the tobacco films by change detection methods. The pixels that change from bare land to film would be regarded as real flue-cured tobacco mulch pixels; the pixels covered by plastic film during both phases would not be tobacco mulch pixels. We also could identify cultivated land from the recent UAV RGB images or collect an existing cultivated land map, and then we could use the cultivated land as a mask to confine the range of identification of tobacco fields. In addition, we could carry other type cameras on UAV, for example, using a thermal camera to differentiate cold buildings and warm tobacco films.

The crops covered by films in the study area include flue-cured tobacco and spring corn. However, the spring corn is covered with white film, while the flue-cured tobacco is covered with black film. The black film and white film can be distinguished from each other through the DN value, and thus the tobacco field and the corn can be easily identified in our study area. However, in other regions, spring cornfields and flue-cured tobacco fields might be covered by the same colour of film, causing misclassifications. Fortunately, the film mulching time of spring corn usually earlier than that of flue-cured tobacco. We could take two phases of UAV images before and after transplantation. The fields covered by the film before transplantation would be spring corn, and the fields covered by film after transplantation would be tobacco.

## 6. Conclusions

Taking the Yunnan Province as an example, as it is the top province for tobacco production in China, in this paper, the best monitoring time for tobacco fields is analysed, a method of extracting the distribution of the flue cured tobacco fields based on an image morphology algorithm is proposed, and the method is verified based on UAV images of 0.04 m GSD. The results showed that the produce accuracy, user accuracy, and overall accuracy of the method proposed in this paper were 92.59%, 96.61% and 95.93%, respectively. The method proposed in this paper has the advantages of automation, flow process, high accuracy and easy operation; however, the GSD of the UAV image had an effect on the accuracy of the proposed method. As a whole, the higher the UAV image GSD, the more suitable it was for mapping the tobacco field distribution by the method proposed in this study.

As we all know, the phenology of tobacco may be different in other study area and land cover varies from area to area. The confusion problems between tobacco and other land cover could be different from the study area. There is no universal approach that can identify tobacco all over the world. In this paper, we provide a research framework for extracting the tobacco planting area. Firstly, analysing the phenological characteristics of flue-cured tobacco and other crops and selecting the best phase for flue-cured tobacco identification. The returning seedling stage after transplantation was beneficial to distinguish flue-cured tobacco from the other crops. Actually, the conclusion is true for major tobacco planting area in China. Second, the plastic films covering tobacco are identified by classification. Finally, adjacent films are merged into film-mulched farmland parcels by image morphology algorithm. The research framework provides a reference for relevant studies (such as mapping mulched farmland and facility agriculture). We also discussed the deficiencies and potential improvements on the proposed tobacco mapping method, which can guild the development of a better identification method of the tobacco field.

For practical applications, we should determine the best monitoring time and the required image GSD by considering many impact factors, including the spatial resolution of the data, the planting structure of the study area, the fragmentation of the farmland, and the phenology calendar of the main crops. Then, we should develop the least costly corresponding identification method to meet the accuracy requirements of tobacco field identification.

## Figures and Tables

**Figure 1 sensors-19-01791-f001:**
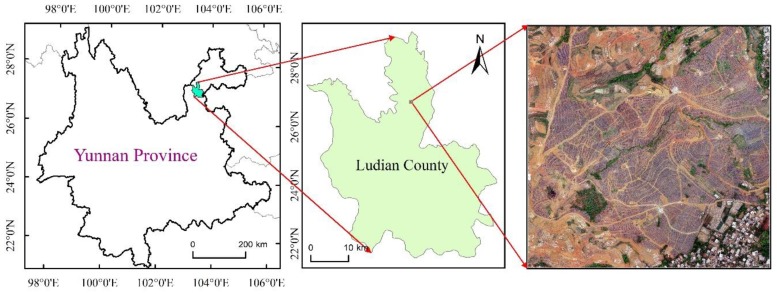
Study area location.

**Figure 2 sensors-19-01791-f002:**
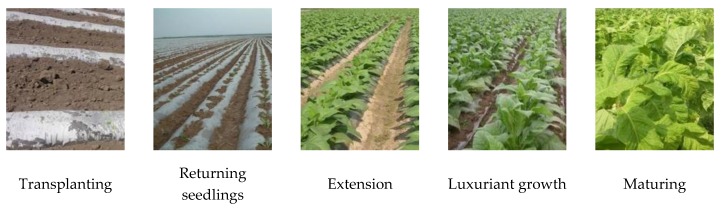
Flue-cured tobacco images during the field period.

**Figure 3 sensors-19-01791-f003:**
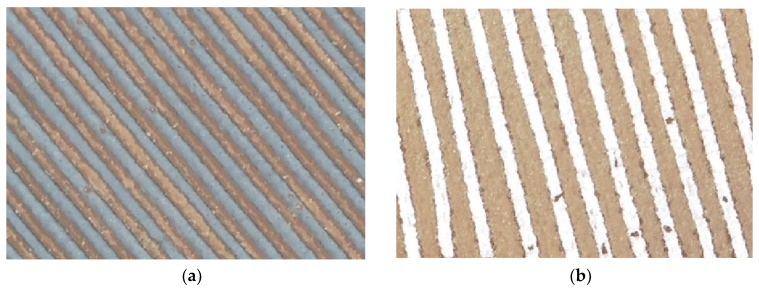
Flue-cured tobacco mulched with black film (**a**) and spring maize mulched with white film (**b**).

**Figure 4 sensors-19-01791-f004:**
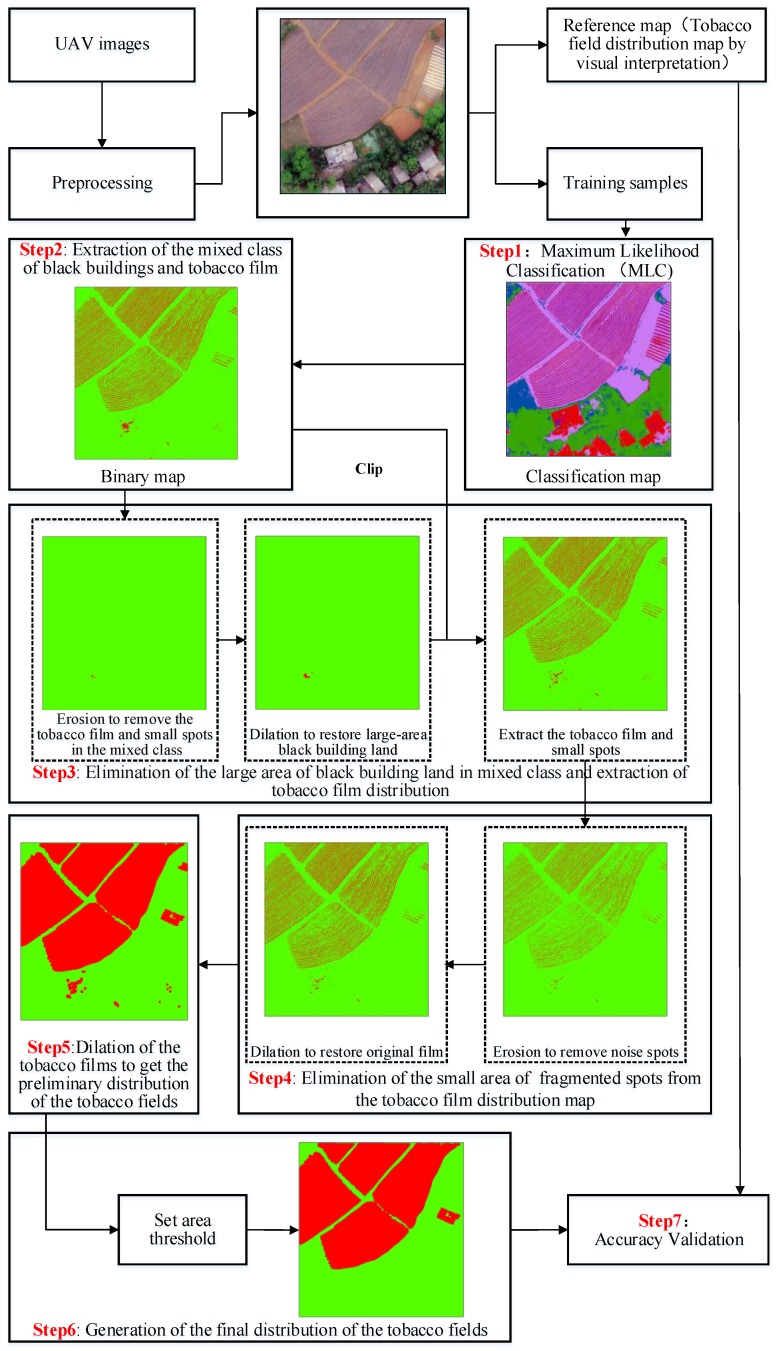
Technology flowchart.

**Figure 5 sensors-19-01791-f005:**
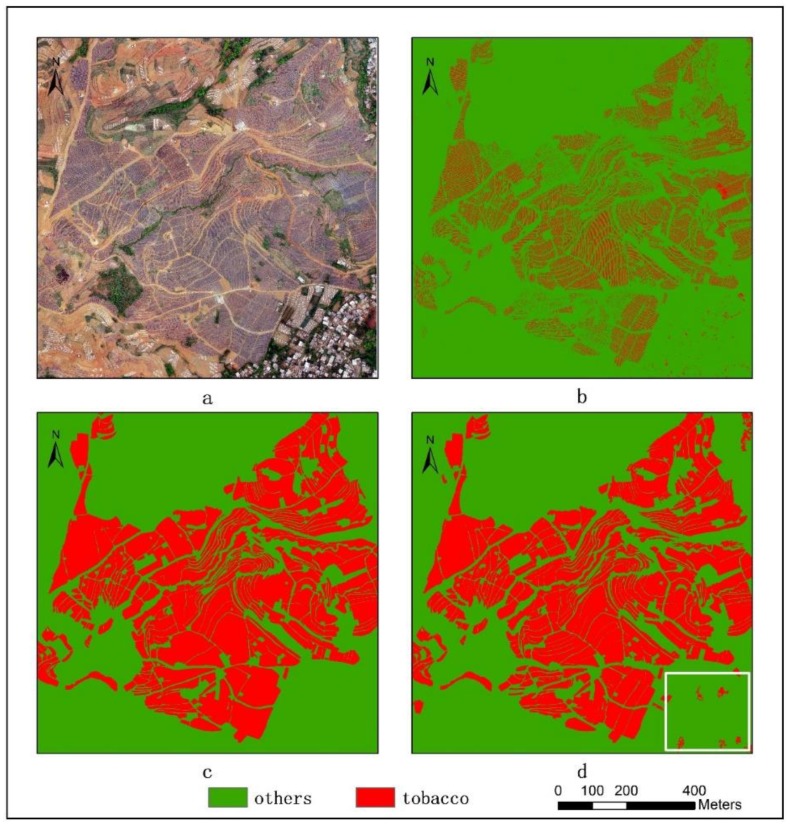
Comparison of the classification results ((**a**) aerial image; (**b**) binary map by recoding the maximum likelihood classification result; (**c**) tobacco distribution map by visual interpretation; (**d**) tobacco distribution map by using the proposed method).

**Figure 6 sensors-19-01791-f006:**
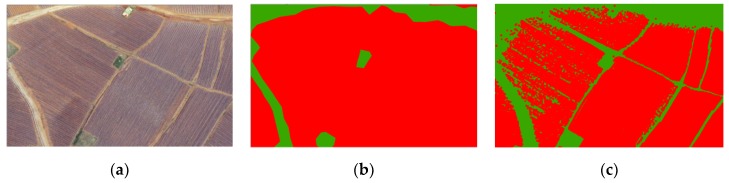
The impact of core size on the dilation operation. (**a**) UAV image; (**b**) dilation with a too large core size; (**c**) dilation with a too small core size.

**Figure 7 sensors-19-01791-f007:**
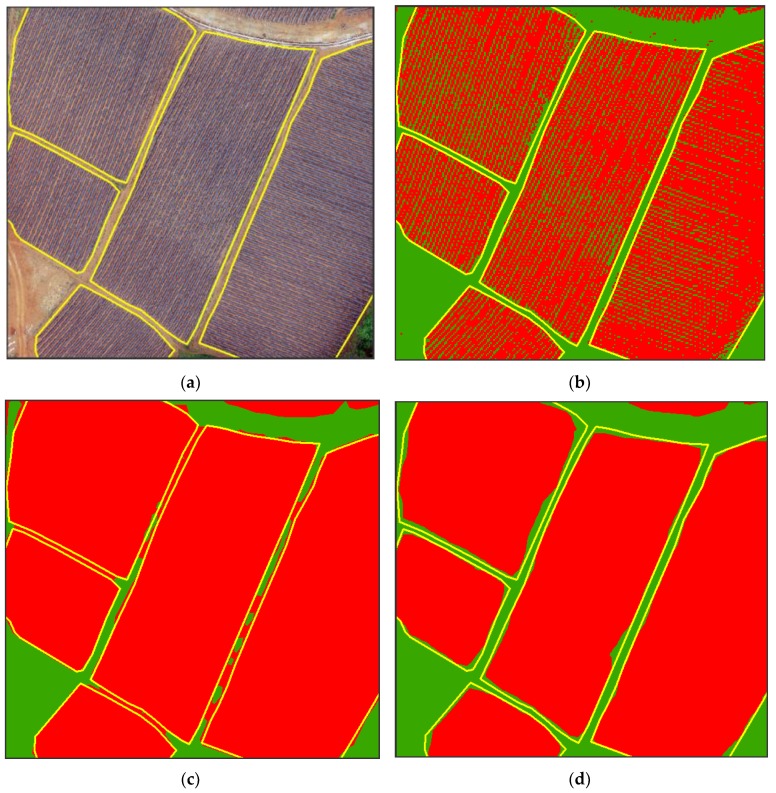
The impact of convolution on tobacco field recognition accuracy. (**a**) Original image; (**b**) binary map; (**c**) morphology; (**d**) combining morphology with convolution.

**Table 1 sensors-19-01791-t001:** The phenological calendar of crops in the study area.

Crop Types		Tobacco	Spring Corn	Summer Corn	Middle Rice
Mar.	Early	Seedlings nurture			
	Middle				
	Late		Sowing (film)		
Apr.	Early				Seedling
	Middle		Three leaf		
	Late	Transplanting			
May	Early		Seven leaf		Transplanting
	Middle				
	Late	Rooting		Sowing	Tilling
Jun.	Early		Jointing	Three leaf	
	Middle	Luxuriant growth			
	Late			Seven leaf	
Jul.	Early		Heading		Jointing
	Middle			Jointing	Booting
	Late	Maturing Harvesting			
Aug.	Early		Maturing	Heading	Heading
	Middle				
	Late				
Sep.	Early		Harvesting	Maturing	Maturing
	Middle				
	Late				
Oct.	Early			Harvesting	Harvesting
	Middle				
	Late				

**Table 2 sensors-19-01791-t002:** Number of training samples for different feature categories.

Categories	Woodland	Grassland	Dirt Road	Facilities Land	Bare Cultivated Land
Training samples quantity (pixel)	1,506,074	280,581	426,194	11,762	243,967
Categories	White building	Black building	White plastic film (tobacco film)	Black plastic film (spring maize film)	
Training samples quantity (pixel)	725,131	34,751	21,630	46,676	

**Table 3 sensors-19-01791-t003:** Accuracy and parameters of tobacco field identification by using different image ground sampling distance (GSD).

GSD (m)	Morphology						Convolution + Morphology			
	Core Size			Accuracy			Window Size	Accuracy		
	*K*	*K*′	*K*″	Produce Accuracy	User Accuracy	Overall Accuracy	S	Produce Accuracy	User Accuracy	Overall Accuracy
0.04	25	13	35	92.59	96.61	95.93	-	-	-	-
0.08	13	7	19	94.07	94.43	95.61	-	-	-	-
0.12	9	5	13	90.05	97.18	95.20	-	-	-	-
0.16	7	3	9	94.23	93.31	95.21	-	-	-	-
0.20	5	-	7	94.30	91.95	94.67	-	-	-	-
0.24	3	-	7	90.97	93.89	94.29	-	-	-	-
0.28	3	-	5	88.81	94.03	93.57	-	-	-	-
0.32	-	-	5	97.19	81.70	90.61	-	-	-	-
0.36	-	-	5	93.15	88.12	92.58	-	-	-	-
0.40	-	-	3	94.39	84.79	91.39	-	-	-	-
0.44	-	-	3	94.61	85.10	91.61	-	-	-	-
0.48	-	-	3	97.33	76.88	87.80	5	81.40	93.77	90.86
0.52	-	-	3	97.99	73.88	86.00	3	84.07	92.91	91.49
0.56	-	-	3	96.83	79.51	89.25	3	76.72	97.21	90.29
0.60	-	-	3	97.73	75.78	87.20	3	80.75	95.21	91.12
0.64	-	-	3	98.05	74.39	86.36	3	81.92	94.98	91.46
0.68	-	-	3	97.77	75.58	87.08	3	79.96	95.89	91.06
0.72	-	-	3	98.11	73.90	86.04	3	81.82	95.49	91.60
0.76	-	-	3	97.66	74.90	86.60	3	80.43	96.16	91.32
0.80	-	-	3	97.77	73.94	85.98	3	80.82	95.71	91.31
0.84	-	-	3	97.99	73.74	85.90	3	82.80	95.77	92.06
0.88	-	-	3	97.64	74.93	86.61	3	79.72	96.76	91.26
0.92	-	-	3	98.13	72.06	84.75	3	82.10	95.78	91.80
0.96	-	-	3	98.12	73.47	85.74	3	81.99	96.79	92.10
1	-	-	3	97.39	74.68	86.39	3	77.69	96.87	90.55

**Table 4 sensors-19-01791-t004:** Examples of mapping the tobacco field distribution by using different methods.

Resolution (m)	Images	Binary Map	Tobacco Field Distribution by Morphology	Tobacco Field Distribution by Combining Morphology with Convolution
0.48	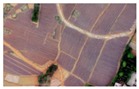	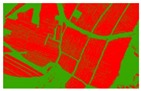	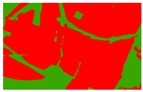	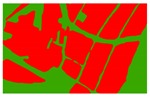
0.60	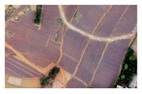	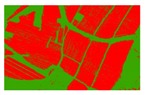	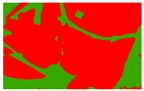	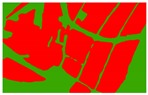
0.76	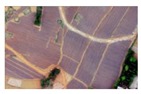	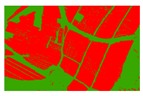	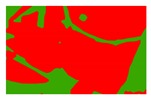	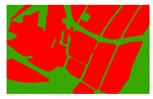
0.88	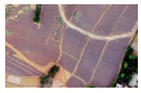	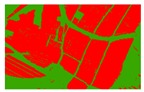	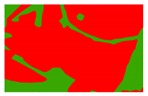	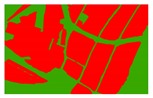
1.00	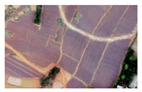	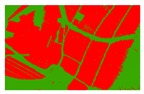	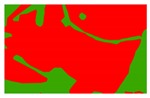	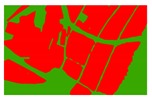

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
