# Peer review of "Mapping Tobacco Fields Using UAV RGB Images"

_sensors, 2019, doi:10.3390/s19081791_

Round 1

Reviewer 1 Report

The author has addressed all my comments. The manuscript has been significantly improved.

minor comment:

Line 266: tobacco film pixels > 50% occupancy leads to 1?

Author Response

Line 266: tobacco film pixels > 50% occupancy leads to 1?

Answer: yes,

After the operation of the median convolution, if the tobacco film pixels occupied a greater area ratio in the window, the value of the pixel in the centre of the window was set as 1; otherwise, it was set as 0. Since the pixel values in the window were only 0 (non- tobacco film) and 1 (tobacco film), if the tobacco film pixels occupied a greater area ratio in the window, the area ratio must be more than 50%. That is tobacco film pixels > 50% occupancy leads to 1.

Reviewer 2 Report

I have compared the previous version with this revised version of this paper. I think the authors have made great efforts to supplement the deficiency according to the comments from different reviewers. The following are some more suggestions for the authors:

(1)Convolution+ Morphology is one of the important issues in this manuscript. However, in the abstract, there is no any description concerning combining convolution. I suggest adding some descriptions about the importance of combining convolution. 

(2)In Figure 6, please revise the typo: UVA image -> UAV image

(3)Please provide the number of images utilized for the orthophoto generation in the manuscript. I thought this information could be helpful for practical authorities.

(4)Please supplement the software or the methods used to produce orthophotos in the manuscript.

(5)The authors have provided the number of training samples. However, I suggest the number of validating samples should also be provided. Besides, I also suggest that the authors may indicate the locations of training and validating samples with points or polygons on the orthophoto map.

Author Response

(1)Convolution+ Morphology is one of the important issues in this manuscript. However, in the abstract, there is no any description concerning combining convolution. I suggest adding some descriptions about the importance of combining convolution.

Answer: revised. We added the following sentence into abstract.

“When the image GSD was reduced to 1 m, the overall accuracy decreased by approximately 10%. To solve this problem, we introduce the convolution method into the proposed method, which can ensure the recognition accuracy of tobacco field is above 90% when GSD is less than or equal to 1 m.”

(2) In Figure 6, please revise the typo: UVA image -> UAV image

Answer: revised.

(3) Please provide the number of images utilized for the orthophoto generation in the manuscript. I thought this information could be helpful for practical authorities.

Answer: there are 400 images utilized for the orthophoto generation. We added this information in line 133-134.

(4) Please supplement the software or the methods used to produce orthophotos in the manuscript.

Answer: Pix4Dmapper. We added this information in line 134-135 in the revised manuscript.

(5) The authors have provided the number of training samples. However, I suggest the number of validating samples should also be provided. Besides, I also suggest that the authors may indicate the locations of training and validating samples with points or polygons on the orthophoto map.

Answer:

The validation data is the tobacco field of whole study area by visual interpretation and digitization. It is already shown in Figure5 (c).

We tied to show the study area with the locations of training samples. However, the polygons of training samples are too small to see. An example is given below.

This manuscript is a resubmission of an earlier submission. The following is a list of the peer review reports and author responses from that submission.

Round 1

Reviewer 1 Report

In this paper, the author presented an automatic identification method for tobacco fields mapping while using UAV-based images. The proposed method is developed by combining supervised classification with image morphological operations. The experimental results demonstrate the feasibility of the proposed approach in providing more than 90% accuracy for the test site. The paper is easy to understand. However, to be honest, the technical contribution of this paper is a little bit limited. Actually, the proposed workflow, which is based on morphological operations after a simple image classification, has been already adopted in various mapping applications. In addition, some important details are missing for the proposed methodology.

Major concerns:

1.     The details for the preprocessing of UAV images are missing. I know the paper is mainly focusing on the image morphological operations. However, the accuracy analysis, such as the accuracy of the derived UAV image-based orthophoto, is still quite important since your following processing (e.g., supervised classification and morphological operations) is directly based on the derived orthophoto.

2.     Please explain if the utilized orthophoto for map generation has been radiometrically corrected. If not, do you think the missing radiometric correction for UAV images may impact the successive supervised classification and the final accuracy of the generated tobacco map?

3.     More details are required for the supervised classification. Since the accuracy of the classification has a significant impact on the final mapping accuracy (the final mapping accuracy can never exceed the achieved accuracy from the supervised classification), the accuracy analysis for the conducted classification has to be provided. In addition, since the final objective is to prepare a map for tobacco fields, a binary classification seems to be sufficient at the classification stage in the proposed method. Please explain the motivation for using 9 categories for the supervised classification. In addition, please explain the reason for the utilization of Maximum Likelihood Classification instead of other methods.

4.     The morphological operations (i.e., erosion and dilation) have been widely used in different applications for raster cleanup purpose. What’s your contribution compared to other research?

5.     Since the proposed morphological method cannot remove all the black buildings, I’m not sure if the proposed workflow for tobacco field mapping is generic enough to deal with different cases. For example, if there is another site with more black buildings, roads, etc., does it mean the proposed method will get a much lower accuracy?

6.     I prefer to consider this paper as a case study for a specific test site instead of a new procedure for solving some general mapping problems. In this regard, I would recommend the author to highlight the contribution of this paper in both abstract and introduction parts.

Other concerns:

1.     Line 109 to 111: Please provide more information regarding the specifications of the utilized platform (e.g., are you using a fixed-wing UAV?). If possible also provide the number of images utilized for the orthophoto generation, as well as some statistics like the percentage of overlap and side-lap of the acquired imagery.

2.     If possible, please provide a figure for the final UAV image-based orthophoto.

3.     Two approaches “corrosion” and “expansion” are introduced in the manuscript. However, it seems “erosion” and “dilation” are the more commonly-used terminology for morphological operation in the research of image processing.

4.     Instead of using image resolution, Ground Sampling Distance (GSD) might be more appropriate to be used in the manuscript.

5.     Section 5 Discussion: Are you using the same training dataset, which was collected on the orthophoto with 0.04m resolution, for all the tests of simulated dataset?

6.     The concept of using the median filter to improve the proposed method is first introduced in Section 5.2. However, both tables 2 and 3, which are presented in Section 5.1, have included the statistics for such convolution-based improvement. This is quite confusing. It is always better to first introduce the concept, and then present the results.

Reviewer 2 Report

The manuscript discusses a methodology for the recognition of tobacco crops using UAV images in an area in China.

The approach is clear and well explained.

The specific methodology of image processing is not new, but its application is original.

The discussion of the results is supported by a good quality of tables and figures.

It can be underlie that, from a spectral point of view, the final classification provides the identification of black films on tobacco crops. This means that, knowing the crops under the film, every crops can be classified with this methodology where this practice is used.

Moreover, it could be interesting to improve this methodology using a thermal camera to differentiate cold buildings and warm tobacco films.

Reviewer 3 Report

The proposed approach has been proved to have the overall accuracy of 84% to 96%. However, 

before providing the proposed approach to the agricultural authorities to calculate the planting areas of flue-cured tobacco, there may be some crucial issues and potential problems which needs to be solved. Please consider the issues listed below:

(1)The planting area of flue-cured tobacco are 1.206 million hectares. However, an UAV image with fine spatial resolution can only cover small area. I think it is hard for the agricultural authorities to collect the UAV data (especially multitemporal data) covered all planting areas. The authors may have to suggest another approach combining multiple sources of remote sensing and in-the-field data with overall consideration.

(2)This paper did not show the map and the range (with longitude and latitude) of the study area. Besides, the covering area of an UAV image did not mention in the manuscript either. Moreover, the phenology (growing period) of tobacco may be different in other Province, County, or even other Village. Last but not least, land cover varies from area to area, and the confusion problems between tobacco and other land cover could be different from the study area. The authors may have to describe more information about the study area, the covering range of UAV image, and discuss the suitability of the proposed approach to the regions outside the study area. 

(3)The criteria of how to select suitable training samples were not described clearly. The authors may have to describe the distribution, size, and number of training samples for tobacco and non-tobacco land cover.

(4)What is the minimum unit for accuracy assessment? Pixel-based or object-based?

(5)What are the spectral bands and their wavelengths of the camera used in the study?